# Integrating GWAS with bulk and single-cell RNA-sequencing reveals a role for *LY86* in the anti-*Candida* host response

Dylan H. de Vries[1], Vasiliki Matzaraki[1,2], Olivier B. Bakker[1], Harm Brugge[1], Harm-Jan Westra[1], Mihai G. Netea[2,3], Lude Franke[1‡]*, Vinod Kumar[1,2‡], Monique G. P. van der Wijst[1‡]*

1 Department of Genetics, University of Groningen, University Medical Center Groningen, Groningen, The Netherlands, 2 Department of Internal Medicine and Radboud Center for Infectious Diseases (RCI), Radboud University Medical Center, Nijmegen, the Netherlands, 3 Human Genomics Laboratory, Craiova University of Medicine and Pharmacy, Craiova, Romania

‡ These authors share last authorship on this work.
* lude@ludesign.nl (LF); m.g.p.van.der.wijst@umcg.nl (MGPVDW)

**Data Availability Statement:** A Seurat object [40], containing the processed single-cell RNA-seq data after QC and cell type assignment, is made

## Abstract

*Candida* bloodstream infection, i.e. candidemia, is the most frequently encountered life-threatening fungal infection worldwide, with mortality rates up to almost 50%. In the majority of candidemia cases, *Candida albicans* is responsible. Worryingly, a global increase in the number of patients who are susceptible to infection (e.g. immunocompromised patients), has led to a rise in the incidence of candidemia in the last few decades. Therefore, a better understanding of the anti-*Candida* host response is essential to overcome this poor prognosis and to lower disease incidence. Here, we integrated genome-wide association studies with bulk and single-cell transcriptomic analyses of immune cells stimulated with *Candida albicans* to further our understanding of the anti-*Candida* host response. We show that differential expression analysis upon *Candida* stimulation in single-cell expression data can reveal the important cell types involved in the host response against *Candida*. This confirmed the known major role of monocytes, but more interestingly, also uncovered an important role for NK cells. Moreover, combining the power of bulk RNA-seq with the high resolution of single-cell RNA-seq data led to the identification of 27 *Candida*-response QTLs and revealed the cell types potentially involved herein. Integration of these response QTLs with a GWAS on candidemia susceptibility uncovered a potential new role for *LY86* in candidemia susceptibility. Finally, experimental follow-up confirmed that *LY86* knockdown results in reduced monocyte migration towards the chemokine MCP-1, thereby implying that this reduced migration may underlie the increased susceptibility to candidemia. Altogether, our integrative systems genetics approach identifies previously unknown mechanisms underlying the immune response to *Candida* infection.

available through the website accompanying our manuscript: https://eqtlgen.org/candida.html. All other relevant data is within the manuscript and Supporting Information files.

**Funding:** M.W. and L.F. are supported by grants from the Dutch Research Council (NWO-Veni 192.029 to M.W. (https://www.nwo.nl/en/), ZonMW-VIDI 917.14.374 to L.F. (https://www.zonmw.nl/en/)), L.F. is supported by a European Research Council Starting Grant (Immrisk 637640 to L.F. (https://erc.europa.eu/)), L.F. is supported by the Oncode institute (https://www.oncode.nl/). V.K. was supported by a grant from the European Society of Clinical Microbiology and Infectious Diseases (2017, https://www.escmid.org/) and a Radboudumc Hypatia Grant (2018, https://www.radboudumc.nl/en/research/academic-and-scientific-training/talent-management/talent-programs/radboudumc-hypatia-track-and-grants/hypatia-tenure-track-grant). MGN was supported by an ERC Advanced Grant (#833247) (https://erc.europa.eu/), and a Spinoza grant of the Netherlands Organization for Scientific Research (https://www.nwo.nl/en/). The funders had no role in study design, data collection and analysis, decision to publish, or preparation of the manuscript.

**Competing interests:** The authors have declared that no competing interests exist.

## Author summary

*Candida albicans* is a fungus that can cause a life-threatening infection in individuals with an impaired immune system. To improve the prognosis and treatment of patients with such an infection, a better understanding of an individual's immune response against *Candida* is required. However, small patient group sizes have limited our ability to gain such understanding. Here we show that integrating many different data layers can improve the sensitivity to detect the effects of genetics on the response to *Candida* infection and the roles different immune cell types have herein. Using this approach, we were able to prioritize genes that are associated with an increased risk of developing systemic *Candida* infections. We expand on the gene with the strongest risk association, *LY86*, and describe a potential mechanism through which this gene affects the immune response against *Candida* infection. Through experimental follow-up, we provided additional insights into how this gene is associated with an increased risk to develop a *Candida* infection. We expect that our approach can be generalized to other infectious diseases for which small patient group sizes have restricted our ability to unravel the disease mechanism in more detail. This will provide new opportunities to identify treatment targets in the future.

## Introduction

*Candida albicans* (*C. albicans*) is an opportunistic fungus colonizing the skin and/or mucosae of approximately 70% of the population [1]. Disruption of the mucosal barrier or a compromised immune system of the host can increase susceptibility to *Candida* infections. This makes it the most common cause of hospital-acquired invasive fungal infections globally [2], with high mortality rates between 33% and 46% [3,4]. The most common form of invasive candidiasis occurs in the blood, known as candidemia [2]. Despite the severity of candidemia and its accompanying research interest, the ability to improve the outcomes for affected individuals has stagnated in recent years. Adjuvant immunotherapy has been suggested as an important strategy to improve patient outcomes, but to implement this a better understanding of the immune response to *Candida* is required [5,6]. As genetics have a great impact on an individual's immune response [7], knowledge on its impact to the anti-*Candida* response will be important as well for the implementation of such therapies.

Genome-wide association studies (GWAS), linking genetic variants to disease risk, have been a commonly used approach to increase disease understanding. However, in the context of candidemia and other infectious diseases, conducting a GWAS is challenging due to the limited size of patient cohorts [8]. Moreover, GWAS studies provide limited insight into the underlying biology of how these genetic variants are linked to *Candida* infection susceptibility. Thus additional approaches are required.

Integrative strategies that combine different molecular datasets in the context of *Candida* infection have been suggested as alternative approaches to prioritize cell types, genes and pathways. These can then be used for follow-up functional studies to better understand candidemia susceptibility. For instance, Smeekens et al. integrated gene expression array data of *Candida*-stimulated PBMCs with genetic information and cytokine measurements from both healthy volunteers and patients with increased susceptibility to *Candida* infections [9]. Using this integrative approach, they identified the interferon pathway as being a crucial host response pathway against *Candida* infection. In a follow-up study, the additive value of integrating multiple molecular datasets became even more apparent as suggestive genetic associations together

with transcriptomic data could prioritize novel pathways implicated in candidemia susceptibility, including the complement and hemostasis pathways [10].

However, further integration is required to understand the mechanism through which genetic variants lead to increased candidemia-susceptibility. These disease-associated variants can be linked to effects on gene expression levels through so-called expression quantitative trait loci (eQTL) analysis. Since disease-associated genetic variants are often regulated in a context-specific manner [11], such eQTL analyses should be performed in such a way that the context-specific nature, i.e. pathogen- and cell type-specificity, can be revealed. With the advent of single-cell RNA-sequencing (scRNA-seq) it now becomes possible to profile the expression of tens of thousands of individual cells at the same time in an unbiased manner [12]. This now allows capturing the context-specific nature of disease-associated genetic variants with increased resolution, while retaining the intercellular dynamics.

Here, we used an integrative approach combining GWAS with bulk and scRNA-seq transcriptomic analyses on *Candida*-stimulated and RPMI control peripheral blood mononuclear cells (PBMCs). By leveraging the sensitivity of bulk RNA-seq data with the context-specific information acquired from scRNA-seq, this integrative approach further improves our understanding of the host response against *Candida*.

## Results and discussion

### Cell type-specific transcriptional response to *Candida albicans*

To reveal the cell type-specific immune response against *Candida*, scRNA-seq analysis was performed on PBMCs from 6 individuals that were stimulated with *Candida* or RPMI control for 24h. After QC, a total of 15,085 cells remained, of which 7,925 cells were RPMI control and 7,160 cells were *Candida*-stimulated. These cells were classified as one of the following six immune cell types: B cells, CD4+ T cells, CD8+ T cells, monocytes, natural killer (NK) cells or plasmacytoid dendritic cells (pDC).

As pathogen-stimulation can potentially affect the cellular state or induce active recruitment of specific cell types, we first determined whether *Candida*-stimulation affected the relative abundance of immune cell types. At baseline, the largest differences in relative abundance of individual cell types varied between 1.6-fold for the CD4+ T cells up to 8.3-fold for the CD8+ T cells (S1 Fig). However, upon stimulation these abundances remained constant within an individual. Overall, CD4+ T cells were the most abundant cell type (61.2%), whereas pDCs were observed the least (1.3%) (Table 1). Even though changes in relative abundances were not detected, we cannot exclude that this is not happening *in vivo*, as our *in vitro* stimulation of PBMCs does not allow detection of active recruitment. Active recruitment of monocytes towards the lymph nodes is part of the host immune response towards *Candida*, as recently shown in mice [13].

Secondly, we identified differentially expressed (DE) genes upon stimulation per cell type separately as well as in all cells together (bulk-like) by performing DE analysis with MAST [14]. This analysis identified a total of 2,384 DE genes in the individual cell types and 3,568 DE genes in the bulk-like sample (Table 1, S1 Table). However, the noisiness and sparseness of single-cell data could potentially introduce artifacts in the DE analysis, resulting in false-positives [15]. To determine the extent to which this occurs, we compared the DE genes identified in the scRNA-seq data with their differential response in a previously described bulk RNA-seq dataset generated from *Candida*-stimulated PBMCs isolated from 70 individuals [7]. This comparison showed that 97.3% of the DE genes from the bulk-like scRNA-seq sample (Fig 1A-I) and at least 96.8% of the DE genes from the individual cell types (Fig 1A-II-VII) could be replicated in the bulk-RNA seq data (S1 Table). Thus, the DE genes identified in scRNA-

**Table 1. Differentially expressed genes per cell type within PBMC single-cell RNA-seq data.**

| Cell type | # cells | # DE genes | # up-regulated | # down-regulated |
|---|---|---|---|---|
| CD4+ T | 9,236 | 1,459 | 1,095 | 364 |
| CD8+ T | 2,300 | 453 | 334 | 119 |
| NK | 1,807 | 1,313 | 927 | 386 |
| B | 789 | 392 | 329 | 63 |
| Monocyte | 757 | 767 | 418 | 349 |
| pDC | 196 | 56 | 49 | 7 |

DE, differentially expressed; NK, natural killer cell; pDC, plasmacytoid dendritic cell; PBMC, peripheral blood mononuclear cell.

seq data reflect true biology rather than artifacts and can be used to uncover the cell type-specific immune response against *Candida*. However, please note that during this prolonged incubation of 24h, it is not possible to distinguish between direct and indirect responses upon *Candida* stimulation.

## *Candida* induces large gene expression differences in CD4+ T cells, NK cells and monocytes

Continuing with the 2,384 DE genes identified in the individual cell types (Fig 1B, S2 Table), we found that 71% of these genes are being upregulated upon stimulation. The majority of these DE genes (1,364) are only found in one cell type, of which the largest part in CD4$^+$ T cells, NK cells and monocytes (558, 468 and 304 DE genes, respectively). The remaining three cell types have very few uniquely identified DE genes, with 27, 5 and 2 DE genes for B cells, CD8$^+$ T cells and pDCs, respectively. As the power to detect DE genes for a cell type is strongly correlated with the number of cells for that particular cell type (Pearson correlation = 0.71) (Panel A in S2 Fig), part of these differences can be attributed to differences in cell numbers (Table 1). However, even when taking this into account, a disproportionately large number of DE genes are specifically identified in the monocytes and NK cells (Panel B in S2 Fig).

To follow-up on these findings, we determined whether the connectivity between each of the major cell types changed upon stimulation with *Candida*. For this, we calculated for each cell type their potential to interact with cells from the same or another cell type by analyzing the expression of cell type-specific receptor and ligand pairs per condition (*Candida*-stimulated and RPMI control), using the computational framework CellPhoneDB [16]. This analysis revealed that especially the B cells (on average 1.67-fold increase) and NK cells (on average 1.62-fold increase) gain additional potential cell-cell interactions upon stimulation with *Candida* (Fig 1C, S3 Table).

Previous studies have reached a consensus that monocytes play an important role in candidemia [17,18], but the contribution of NK cells is less clear [19,20]. Interestingly, specifically in immunocompromised mice the depletion of NK cells increased the susceptibility to candidemia [21]. As in humans, candidemia mainly affects immunocompromised patients, we hypothesize that NK cells are likely to play an important role in the human candidemia response as well. Through the DE and ligand-receptor expression analysis we show that, in addition to monocytes, also NK cells are strongly activated and are increasingly connected to other cells. This provides extra evidence for their importance in the immune response against *Candida*.

In addition to these unique responses, 1,020 DE genes were identified across multiple cell types, of which a core of 41 DE genes was shared between all six cell types. Of these shared DE genes, 89.8% of effects have the same direction across all responding cell types (Fig 1D).

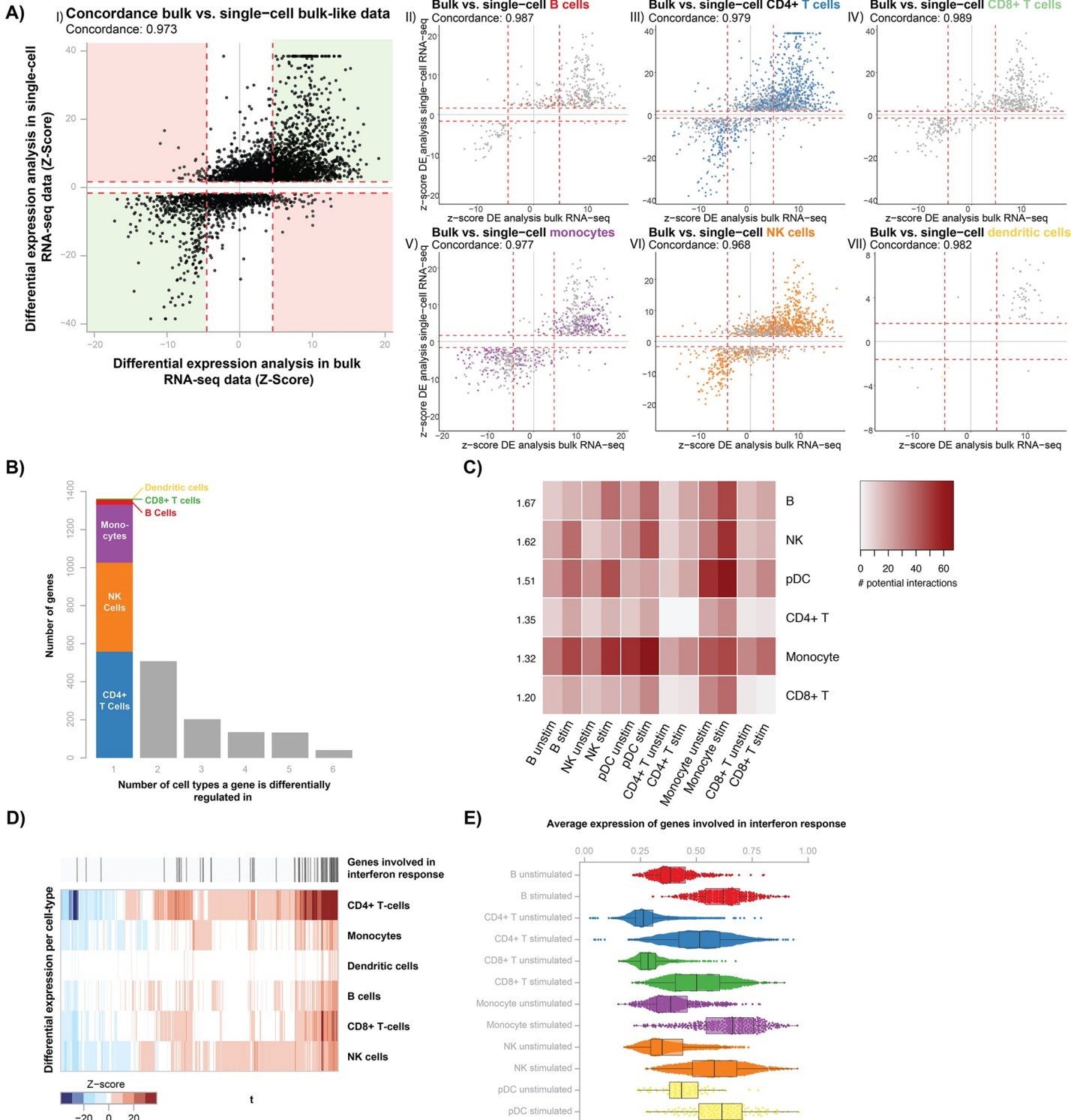

**Fig 1. Single-cell RNA-seq differential expression analysis reveals the cell type-specific response to *Candida* stimulation.** (A) Comparison of differentially expressed (DE) genes upon *Candida* stimulation identified in 6 individuals for whom single-cell RNA-seq (scRNA-seq) data is generated (y-axis) as compared to the effect in 70 bulk RNA-seq samples (x-axis). Each dot represents a DE gene and the dotted red lines indicate the significance thresholds. In panel I (DE genes in bulk-like scRNA-seq sample, which contains all cells from an individual), concordant DE genes are shown in the green area and discordant genes in the red area. In panels II-VII (DE genes in specific cell type), color indicates whether a DE gene is cell type-specific. (B) Bar plot showing the sharedness of DE genes across cell types. The first bar, with cell type-specific DE genes, is colored based on the cell type in which the DE gene is found. (C) Heatmap of the total number of ligand-receptor interactions between cells of

the same or different cell types. Each cell type is compared to cell types of the same condition (RPMI control left, 24h *Candida*-stimulation right). Each row has a number showing the average fold enrichment in ligand-receptor pair interactions between that cell type and all cell types. (D) Heatmap of DE gene Z-scores per cell type (y-axis) for genes that are identified as DE in more than one cell type (x-axis). Red colors indicate upregulation and blue colors show downregulation upon Candida stimulation. Above the heatmap, genes found within the interferon pathway are highlighted. (E) Box plots showing the mean expression of interferon pathway-associated genes (x-axis) for each cell type and stimulation condition (y-axis). Box plots show the median, first and third quartiles, and 1.5× the interquartile range and each dot represents the expression of a single cell.

Moreover, these shared DE genes showed the strongest differential effect upon stimulation (Fig 1D). Pathway analysis on the core set of 41 DE genes revealed strong enrichment of the interferon pathway (P = $10^{-22}$) (S2 Table). This is in line with previous findings in PBMC bulk expression data that showed strong differential expression of the interferon pathway upon 24h *Candida* stimulation [9]. Notably, when taking the average expression of all interferon pathway-associated genes per cell, the strength of upregulation of the interferon I pathway after *Candida* stimulation is consistent across all cell types (Fig 1E).

## Identification of *Candida*-response QTLs using bulk RNA sequencing

In addition to identifying cell type-specific responses to *Candida* infection, we also studied the effect of genetic variants on gene expression levels before and after *Candida* stimulation using previously published bulk RNA-seq data from PBMCs [7]. The rather small sample size of this study limits its predictive power, in part by the large multiple testing burden of genome-wide eQTL analysis [22]. To reduce the multiple testing burden, we limited our single nucleotide polymorphism (SNP)-gene combinations to only the 16,990 top *cis* SNP-gene pairs identified in the largest eQTL meta-analysis to date [23], containing unstimulated whole blood samples of 31,684 individuals. However, by confining our analysis only to previously reported *cis*-eQTLs in unstimulated blood samples, we might miss out on eQTLs that only show up after stimulation. Nevertheless, if there is a weak effect without stimulation that is strengthened by *Candida* stimulation, restricting ourselves to previously identified top SNP-gene pairs will increase our chance to detect the eQTL effect. Using this approach, a total of 1,563 and 1,637 eQTLs were found in 72 *Candida*-stimulated and 75 RPMI control samples, respectively (Fig 2A, S4 Table). Whilst many (44%) of these eQTLs were found both before and after stimulation, the majority of eQTLs were condition-specific (Fig 2A). By subtracting per individual and per gene the *Candida*-stimulated expression from the RPMI control expression, we also tested whether certain SNPs affected the expression of a particular gene with different effect sizes before and after stimulation. This so-called response QTL analysis was performed in the 67 individuals for which both *Candida*-stimulated and RPMI control conditions were assessed and revealed 27 response QTLs (S4 Table). Subsequently, scRNA-seq data was used to pinpoint the potential cell type in which the response QTL effects manifest themselves (S3 Fig). Annotation of the cell type- and context-specificity of eQTLs may help to understand their involvement in human disease.

## Prioritization of *LY86* as a potential key driver gene for candidemia

Previously, it was shown that integrating multiple molecular datasets can help prioritize disease-relevant genes, cell types and pathways [9,10]. Therefore, as a next step, we took the GWAS summary statistics of a previously published candidemia cohort of 161 cases and 152 disease-matched controls [8] and overlaid this with our 27 response QTLs (Fig 2B). This revealed an enrichment of candidemia-susceptibility SNPs within the *Candida*-response QTL SNPs ($\lambda_{inflation}$ = 1.49) (Fig 2C). The top enriched response QTL SNP was rs9405943 (P = 1.2 x $10^{-3}$, OR = 0.594), and was in near perfect linkage disequilibrium with rs2103635, the SNP

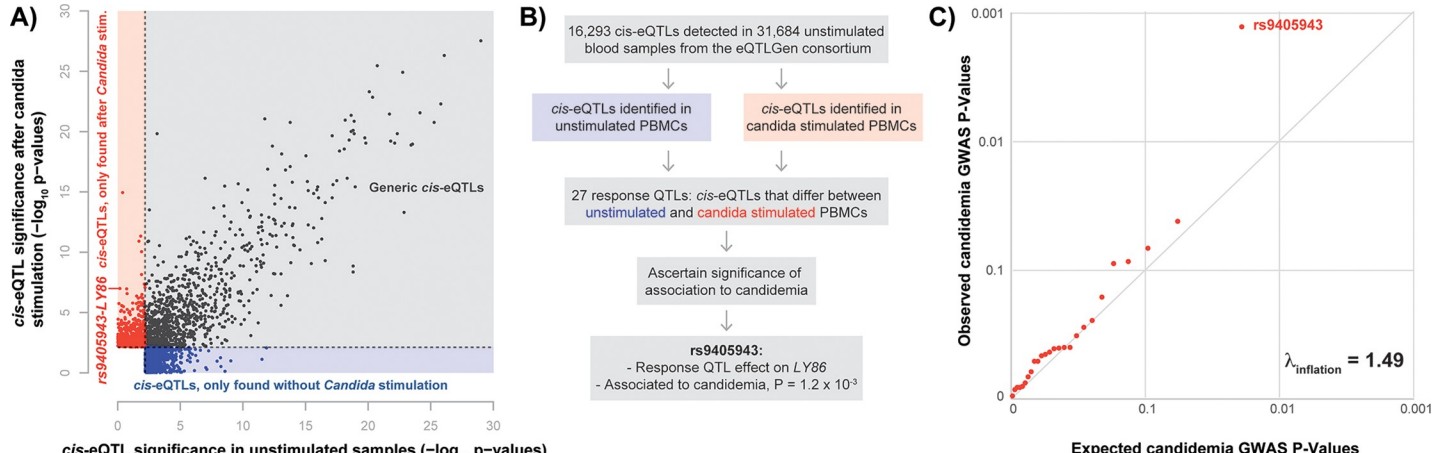

**Fig 2. Integration of GWAS with eQTL analysis allows for prioritization of potential key driver genes.** (A) Comparison of -log10 p-values of identified eQTLs in individuals without *Candida* stimulation (n = 75) and eQTLs identified in individuals after *Candida* stimulation (n = 72). Red points show eQTLs that are significant only with *Candida* stimulation, blue points show eQTLs that are significant only without *Candida* stimulation and black points depict eQTLs that are significant in both conditions. The eQTL analysis was restricted to top SNP-gene combinations identified in the eQTLGen consortium [23]. (B) The performed work flow to identify potential key driver genes in Candida response. (C) QQ-plot of 27 *Candida*-response QTL SNPs in a GWAS on candidemia susceptibility, comparing expected GWAS P-values (x-axis) with observed GWAS P-values (y-axis). The dots show deviation from the expected line ($\lambda_{inflation}$ = 1.49) with the strongest GWAS association found for rs9405943.

showing the strongest association with candidemia in this region (P = 7 x $10^{-4}$, OR = 0.58) ($r^2$ = 0.94) (S4 Fig). SNP rs9405943 showed a strong effect on expression levels of *LY86* after *Candida* stimulation (β = 0.58, P = 1.5 x $10^{-7}$), but not in the RPMI control condition (β = 0.05, P = 0.68) (Fig 3A).

The expression of *LY86* is strongly downregulated upon *Candida* stimulation in the bulk RNA-seq dataset (P = 7.2 x $10^{-28}$). Additionally, we see that the candidemia-risk allele *A* at rs9405943 is associated with stronger downregulation of *LY86* after stimulation. This suggests that high expression of *LY86* has a protective function against candidemia. Single-cell gene expression data shows that both B-cells and monocytes express *LY86*. However, only expression in monocytes is affected by the stimulation (P = 1.9 x $10^{-14}$) (Fig 3B and 3C), suggesting that this gene contributes to candidemia susceptibility through monocytes.

It is known that LY86 forms a complex with Toll-like receptor protein RP105 and is involved in several immune disorders [24–26]. Depending on the cell type, this complex has opposite regulatory effects on TLR4 signaling [27,28]; while TLR4 signaling is activated and stimulates proliferation and antibody production in B-cells, it is negatively regulated in myeloid cells. These opposite effects likely reflect the engagement of different cell type-specific co-receptors [28]. While previous studies have shown the importance of the RP105/LY86 complex in mediating the TLR4-mediated innate immune response against bacterial lipopolysaccharides (LPS) [29,30], its role in the anti-*Candida* response is unknown.

In monocytes, both increased signaling activity of TLR4 and absence of RP105 are associated with downregulation of the chemokine receptor CCR2, leading to their reduced migratory capacity [25,31]. Through complex formation with LY86, RP105 inhibits TLR4 signaling in monocytes [28]. Therefore, we hypothesize that the rs9405943 candidemia-risk allele *A*, which lowers *LY86* expression in monocytes upon *Candida* stimulation, will decrease the migratory capacity of monocytes, which ultimately increases susceptibility to candidemia (Fig 3D). In line with this, in mice the trafficking of CCR2-dependent inflammatory monocytes has been shown to play an essential role in fungal clearance during systemic candidiasis in the first 48h of infection [18].

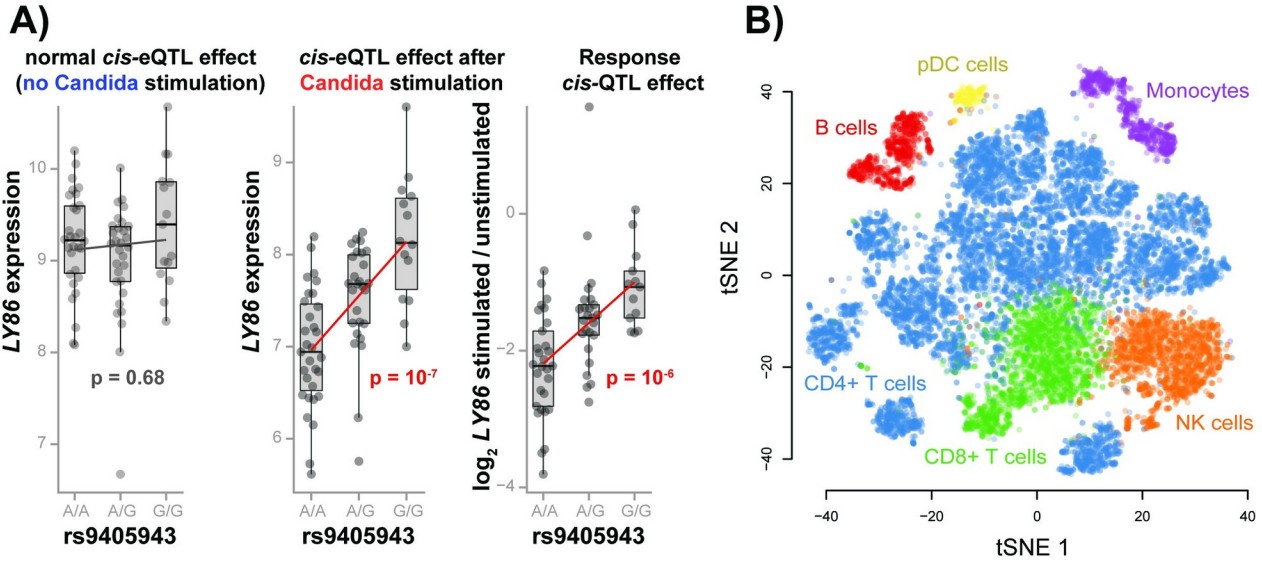

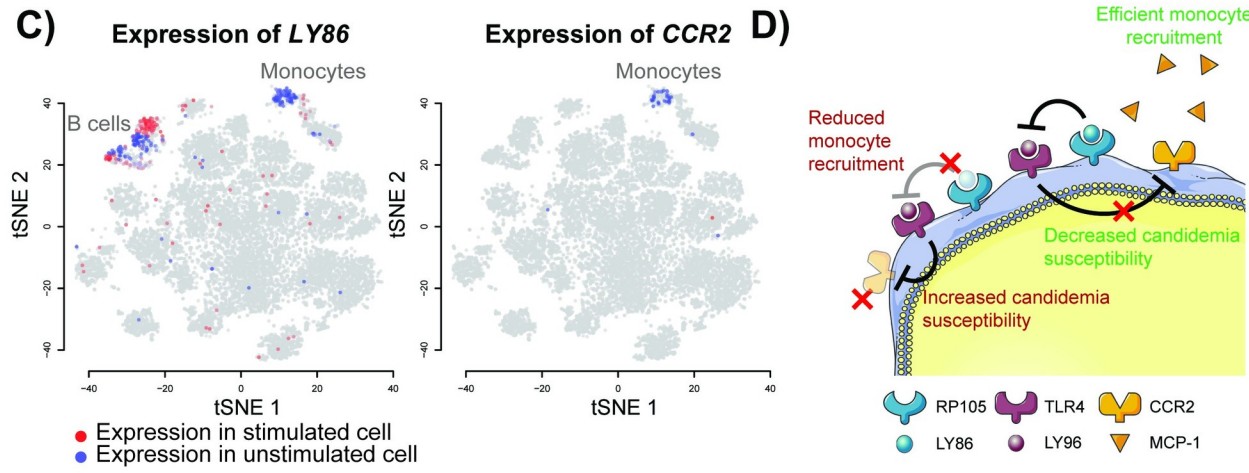

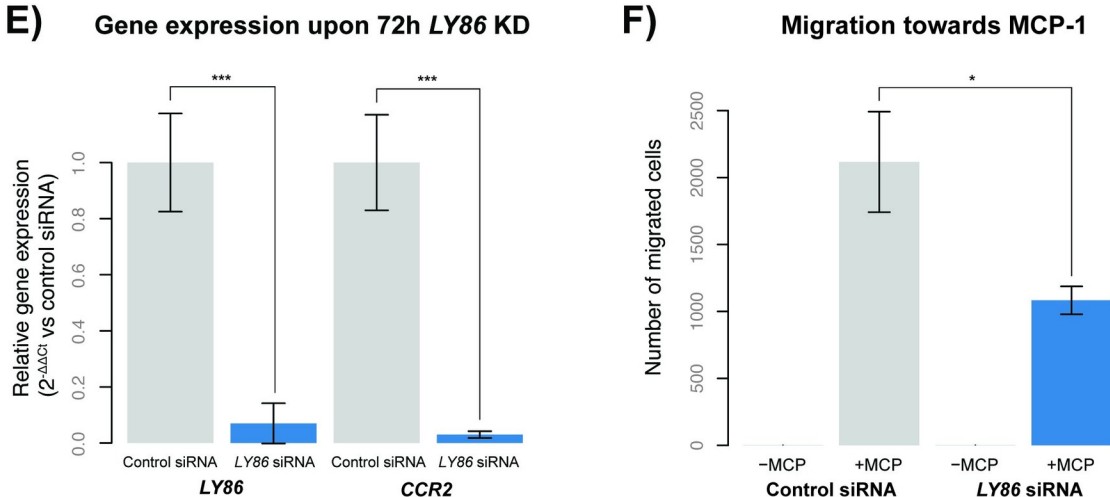

**Fig 3. Proposed mechanism of *LY86* in candidemia susceptibility.** (A) Box plots showing the effect of rs9405943 genotype on *LY86* expression without *Candida* stimulation (left), after *Candida* stimulation (center) and the response difference to *Candida* stimulation (right), as calculated in bulk RNA-seq data. Box plots show the median, first and third quartiles, and $1.5 \times$ the interquartile range and each dot represents the expression of an individual. The x-axis shows the rs9405943 genotype and the y-axis shows the expression or expression response difference for *LY86*. Red p-values indicate significant effects. (B) A tSNE plot generated with single-cell expression data with and without *Candida* stimulation, colored by cell type. (C) Two tSNE plots colored by the expression of *LY86* (left) and *CCR2* (right). Red cells indicate expression in a stimulated cell, blue cells indicate expression in an unstimulated cell and gray cells have no expression. (D) The proposed working mechanism for LY86 on candidemia susceptibility in monocytes. LY86 (aka MD-1) can form a complex with RP105 (aka CD180) and this complex directly binds to the LY96(aka MD-2)-TLR4 complex, thereby inhibiting TLR4 signaling. However, in individuals with the rs9405943 candidemia-risk allele, LY86-RP105 complex formation is reduced, and as a result, LY96-TLR4 signaling is increased. As a consequence, TLR4-mediated chemokine receptor 2 (CCR2) repression increases, which reduces monocyte recruitment towards MCP-1 (aka CCL2) and increases candidemia susceptibility. (E) Normalized *LY86* and *CCR2* gene expression levels upon 72h *LY86* siRNA or control siRNA treatment in THP-1 monocytes. Each bar represents the mean ± SD of three independent experiments, *** $p < 0.001$ (F) Migration rate of 72h *LY86* versus control siRNA treated THP-1 cells towards MCP-1 or RPMI medium without serum. Each bar represents the mean ± SD of three independent experiments, *** $p < 0.001$.

Of note, the TLR4 signaling pathway has been shown to be involved in the innate immune responses of several microbial and fungal infections [32–35]. In addition, a previous study, in which PBMCs from 8 individuals were stimulated for 24h with microbial and fungal pathogens, showed reduced expression of *LY86* after stimulation with *Mycobacterium Tuberculosis* (-1.20-fold, $P = 1.53 \times 10^{-7}$), *Borrelia* (-1.34-fold, $P = 7.31 \times 10^{-13}$), *Pseudomonas Aeruginosa* (-1.31-fold, $P = 9.45 \times 10^{-9}$) and *Streptococcus Pseudomoniae* (-1.54-fold, $P = 2.01 \times 10^{-19}$), but not *Aspergillus Fumigatus* (-0.012-fold, $P = 0.98$) [7]. Altogether, this indicates that the differential regulation of *LY86* in monocytes, as seen in response to *Candida*, could also affect the susceptibility towards other blood-based bacterial infections.

## Functional validation of the role of *LY86* in monocytes

To test our hypothesized mechanism of action (Fig 3D), we conducted experimental follow-up studies in THP-1 monocytes. As the candidemia risk allele in combination with *Candida* stimulation is associated with reduced expression of *LY86* specifically in the monocytes (Figs 2C and 3A–3C), we used siRNA knockdown of *LY86* to mimic this effect. 72h after siRNA treatment, we confirmed efficient knockdown of *LY86* (14.3-fold lower expression, $P = 0.001$) by qPCR. In line with our hypothesis, the expression of *CCR2* was also reduced (33.3-fold, $P = 0.0006$) upon knockdown of *LY86* (Fig 3E, S5 Table). These 72h *LY86* or control siRNA treated cells were then used in a migration assay to assess their migratory capacity towards the chemokine MCP-1 or serum-free medium as a control. After 3h incubation, we only observed migration towards MCP-1 and not the serum-free medium. Notably, the migratory capacity towards MCP-1 of the *LY86* siRNA treated cells was reduced (2.0-fold, $P = 0.01$) as compared to control siRNA treated cells (Fig 3F, S5 Table). Summarized, these results indicate that reduced expression of *LY86* can reduce the migratory capacity of monocytes, potentially through reduced expression of *CCR2*, and thereby, may increase the susceptibility to candidemia.

## Final discussion and conclusion

In summary, we present an integrative approach of GWAS, bulk RNA-seq and scRNA-seq data to extract important knowledge about candidemia susceptibility. Such an integrative approach is valuable in the context of infectious diseases, such as candidemia, for which the limited size of patient cohorts limits the power of the GWAS. Otherwise, a GWAS alone would require much larger sample sizes in order to extract useful information from such studies. Moreover, a GWAS alone cannot explain how genetic variation affects disease or which cell type will be affected, and therefore, a systematic integration of different molecular datasets may be the only avenue to reveal this information. By combining these data layers, we

corroborate the previously identified importance of the IFN pathway and of monocytes in *Candida* infections [9]. In addition, we provide new evidence for a strong response in NK cells against *Candida* and a potential novel role for the *LY86* gene in candidemia susceptibility.

Our integrative approach is not limited to *Candida* infection, but can also be applied to gain a better insight into other infectious diseases for which the progress of disease understanding is hindered by small patient cohorts. We expect that in the near future, the cell type-specific and context-specific resolution of this integrated approach can be further improved as large-scale scRNA-seq datasets become readily available in many different individuals, stimulation conditions and diseases through large-scale consortia such as the single-cell eQTLGen (https://eqtlgen.org/single-cell.html) [36] and LifeTime consortium (https://lifetime-fetflagship.eu). Such increased resolution would allow reconstruction of personalized, disease-specific gene regulatory networks that could provide us with new insights that could guide new treatment opportunities [37].

## Materials and methods

### PBMC collection and *Candida*-stimulation

Whole blood from 6 individuals of the northern Netherlands population cohort Lifelines Deep [38] was drawn into EDTA-vacutainers (BD). PBMCs were isolated and maintained as previously described [39]. In short, PBMCs were isolated using Cell Preparation Tubes with sodium heparin (BD) and were cryopreserved until use in RPMI 1640 containing 40% FCS and 10% DMSO. After thawing and a 1h resting period, $50 \times 10^4$ cells were seeded in 200 μl RPMI1640 supplemented with 50 μg/mL gentamicin, 2 mM L-glutamine, and 1 mM pyruvate in a nucleon sphere 96-round bottom well plate. Cells were either stimulated or kept unstimulated for 24h with $1 \times 10^6$ heat-killed *C. albicans* blastoconidia (strain ATCC MYA-3573, UC 820) CFU/ml at 37˚C in a 5% $CO_2$ incubator. After 24h, cells were washed twice in medium supplemented with 0.04% bovine serum albumin. Cells were counted using a haemocytometer and cell viability was assessed by Trypan Blue.

### Single-cell library preparation and sequencing

Three, sex-balanced sample pools were prepared each aimed to contain 1750 cells/donor from 6 donors (10,500 cells). One pool contained only unstimulated cells, one pool only stimulated cells and one pool contained a 50/50 mixture of both. Each sample pool was loaded into a different lane of a 10x chip (Single Cell A Chip Kit, 120236). The 10x Chromium controller (10x Genomics) in combination with v2 reagents was used to capture the single cells and generate sequencing libraries according to the manufacturer's instructions (document CG00026) and as previously described [39]. Sequencing was performed using the Illumina HiSeq 4000 with a 75-bp paired-end kit, performed by GenomeScan (Leiden, the Netherlands).

### Single-cell RNA-seq alignment, preprocessing and QC

Alignment, demultiplexing and cell type classification of the scRNA-seq data was performed as previously described [39], but now using the 2.3.0 version of Seurat [40]. After QC, 15,085 cells remained of which 7,160 were stimulated and 7,925 were unstimulated. The stimulated and unstimulated cells were combined into a single dataset using Canonical Correlation Analysis (CCA) [40], by taking the first 20 dimensions. Clusters were identified using the FindClusters function from Seurat, using the first 20 dimensions in the CCA space. Expression of known marker genes was assessed to assign cell types to each cluster, resulting in the identification of six major cell types.

### Single-cell RNA-seq differential expression analysis

Differential expression (DE) between *Candida*-stimulated and RPMI control cells was calculated for each cell type separately and in a bulk-like analysis using the MAST implementation of the Seurat package [14]. All genes without expression in at least 1 cell were removed, leaving 20,236 genes. Bonferroni multiple testing correction was applied, yielding a significance threshold of 2.47e-06. Genes that were differentially expressed in all cell types (i.e. core genes) and each cell type individually were used as input for the ToppFun functional enrichment analysis using the REACTOME pathway [41]. P-values were calculated using the probability density function and were Bonferroni corrected.

### Cell-to-cell interaction potential analysis

The potential of cell-to-cell communication through ligand/receptor pair interactions was studied using version 2 of CellPhoneDB [16]. This software uses the normalized expression data and the cell type classifications to see which cell types have expression of known ligands and receptors to estimate whether there is an interaction potential between cells of the same or different cell types. The analysis was performed on each cell type and condition (24h *Candida*-stimulated versus RPMI control) separately. CellPhoneDB was run using the default database of ligand-receptor interactions provided with the software and was run using default settings for p-value thresholds (0.05), expression threshold (expression in $> = 10\%$ cells) and permutations (1,000).

### Bulk RNA-seq data on *Candida*-stimulated PBMCs

All bulk RNA-seq data from PBMCs was previously generated [7] in 70 individuals from the GONL cohort [42]. This data was generated from PBMCs that were stimulated for 24h with *Candida* or remained unstimulated (RPMI control condition). Details of the stimulation are similar to the scRNA-seq data on *Candida*-stimulated PBMCs as mentioned above, and have been previously described [9]. The differentially expressed genes upon stimulation were previously identified [7] through DESeq2 [43]. The differential expressed genes identified in the scRNA-seq data were compared with the differential response in this bulk RNA-seq data.

### Bulk RNA-seq eQTL analysis

Of the same bulk RNA-seq cohort, eQTLs were identified in the data from 72 individuals and 75 individuals for *Candida*-stimulated and RPMI control conditions, respectively. The response QTLs were identified in the 67 individuals for which both conditions were assessed and genotype information is available. To calculate this, we subtracted per individual and per gene the *Candida*-stimulated expression from the RPMI control expression and tested whether certain SNPs affected the expression of a particular gene with different effect sizes before and after stimulation. All expression data were log2 transformed before being used in the 1.2.4F version of the QTL pipeline as described before [44]. To reduce the multiple testing burden, analysis was restricted to the list of 16,989 top SNP-gene combinations identified in the largest whole blood eQTL meta-analysis to date containing 31,684 whole blood samples [23]. This list of top SNP-gene combinations contains SNPs with minor allele frequencies (MAF) >0.01, Hardy-Weinberg P-values >0.0001, call rate >0.95, and MACH r2 > 0.5 within a 1Mb window of the gene. An FDR threshold of 0.05 was used as significance cut-off, using the permutation strategy described in Westra et al. with 100 permutations [45].

## GWAS on candidemia susceptibility

The GWAS on candidemia susceptibility was previously described [8]. In short, this GWAS was performed in a cohort of 161 candidemia cases and 152 disease-matched controls of European ancestry whose demographic and clinical characteristics have been previously described [46]. DNA was genotyped using Illumina HumanCoreExome-12 v1.0 and HumanCoreExome-24 v1.0 BeadChip SNP chips. Genotypes were imputed using the human reference consortium reference panel [47] using the Michigan imputation server [48]. In total, 5,426,313 SNPs were tested for disease association using Fisher's exact test with PLINK v1.9 [49]. The lambda inflation was calculated by taking the GWAS p-values for each of the 27 response-QTL SNPs, regardless of whether the GWAS p-value was significant.

## siRNA treatment

Before starting the experiment, THP-1 monocytes were maintained in RPMI medium supplemented with 10% FBS and 1% Pen-Strep at 37˚C in a humidified 5% $CO_2$ incubator. 50,000 THP-1 monocytes were seeded in round-bottom 96-wells plates. During seeding, 1 μM Accell human *LY86* siRNA SMARTpool (Dharmacon) or 1 μM Accell Green non-targeting siRNA control (Dharmacon) was delivered to these cells in 100 ul Accell delivery medium. After 24h, this procedure was repeated by adding an additional 100 ul to each well. After 72h, *LY86* and *CCR2* mRNA levels were quantified using qRT-PCR and migration rate was quantified using a migration assay.

## Quantitative real-time PCR (qRT-PCR)

RNA was isolated using QIAzol lysis reagent according to manufacturer's instructions. RNA was quantified using a Nanodrop 1000 spectrophotometer (Thermo Scientific). 400 ng RNA was reverse transcribed into cDNA using random hexamer primers with the RevertAid H Minus First Strand cDNA Synthesis Kit (Thermo Scientific) following manufacturer's protocol. Each qRT-PCR reaction contained 500 nM of each primer pair (Table 2), 10 ng of cDNA and 1x iTaq universal SYBR green supermix (Bio-Rad). qRT-PCR reactions were conducted on the Quantstudio 7 Flex real time PCR (Thermo Fischer) for 10 min at 95 ˚C, followed by 40 cycles of 15 sec at 95 ˚C and 30 sec at 60 ˚C. GAPDH was used as housekeeping gene. Data and melting curves were analyzed using Quantstudio Real-time PCR software v1.3 and relative expression compared to controls was calculated using the ΔΔCt method [50]. Significance was calculated using an unpaired t-test.

## Migration assay

The Boyden Chamber transwell migration assay was used to determine the migration rate towards MCP-1 upon *LY86* KD [51]. A polycarbonate membrane insert with a 5 μM pore size (Cell Biolabs) was placed in a well of a 24-wells plate filled with 500 μl Accell delivery medium supplemented with 0.5% BSA with or without 100 ng/ml human MCP-1 (Prospec). The insert was filled with 100 μl Accell delivery medium supplemented with 0.5% BSA and 100,000 THP-1 monocytes treated for 72h with *LY68* siRNA or Green non-targeting siRNA. Cells were placed in a humidified incubator with 5% CO2 at 37 ˚C. After 3h, the number of migratory cells was quantified in the bottom well using a hemocytometer. Significance was calculated using an unpaired t-test.

**Table 2. qRT-PCR primer sequences.**

| Target gene | Forward primer (5'-3') | Reverse primer (5'-3') |
|---|---|---|
| *LY68* | TGTGGAAGAAGGAAAGGAGAGCA | GTACAGTTCCAGCAAAACCTGG |
| *CCR2* | AGTTGCTGAGAAGCCTGACA | TCTCTGTTCAGCTTGTGGCT |
| *GAPDH* | CCACATCGCTCAGACACCAT | GCGCCCAATACGACCAAAT |

## Code availability

The original R code for Seurat [40] (https://github.com/satijalab/seurat), CellPhoneDB v2.0 [16] (https://github.com/Teichlab/cellphonedb) and our in-house eQTL pipeline [44] (https://github.com/molgenis/ systemsgenetics/tree/master/eqtl-mapping-pipeline) can be found at Github. All custom-made code is made available via GitHub (https://github.com/molgenis/scRNA-seq).

## Ethics statement

The LifeLines DEEP study was approved by the ethics committee of the University Medical Center Groningen, document number METC UMCG LLDEEP: M12.113965. All participants signed an informed consent form before study enrollment. All procedures performed in studies involving human participants were in accordance with the ethical standards of the institutional and/or national research committee and with the 1964 Helsinki declaration and its later amendments or comparable ethical standards.

## Supporting information

**S1 Fig. Relative cell type frequencies per donor.** The relative cell type frequency per donor (A) before and (B) after 24h *Candida* stimulation for each of the 6 cell types identified within the peripheral blood mononuclear cells: B-cells, CD4+ and CD8+ T cells, monocytes, natural killer (NK) cells and plasmacytoid dendritic cells (pDCs).
(TIF)

**S2 Fig. Differentially expressed genes per cell type.** The number of differentially expressed (DE) genes per cell type (A) for all and (B) for the unique DE genes against the number of cells.
(TIF)

**S3 Fig. Bulk *Candida*-response QTLs visualized in single-cell RNA-seq data.** 27 response QTLs were identified in PBMC bulk RNA-seq data of 24h *Candida*-stimulated cells compared to RPMI control cells. To pinpoint the cell type in which the response QTL effect could manifest itself, PBMC single-cell RNA-seq data of 24h *Candida*-stimulated cells compared to RPMI control cells was used. For 21 out of the 27 response QTL genes, expression was detected in the single-cell RNA-seq data. The expression of individual cells was colored according to the condition: red for stimulated and blue for RPMI control cells. The expression level is colored by intensity, with gray cells having no expression.
(TIF)

**S4 Fig. Regional association plot of the LY86 locus.** Regional association plot of the LY86 locus (chr6;6088933–7155216, build hg19, ± 500 kb). P values on the -log10 scale are presented as a function of the chromosomal position. The top enriched *Candida*-response QTL SNP rs9405943 is presented as the red circle and the SNP rs2103635 showing the strongest association to candidemia susceptibility is shown as the purple diamond. The correlations ($r^2$) of each

of the surrounding SNPs to SNP rs2103635 are shown in the indicated colors. Recombination rate is shown in pale blue.
(PDF)

**S1 Table. Differential expression analysis upon *Candida* stimulation in single-cell as compared to bulk RNA-seq data.** Differentially expressed (DE) genes in the single-cell RNA-seq (scRNA-seq) data of 24h *Candida*-stimulated cells as compared to RPMI control cells per cell type separately and for all cells together (bulk-like). The first tab provides an overview of the 41 core genes that are DE in all cell types, including the Bonferroni-corrected p-value in each of the cell types (p_val_adj). The p-value (p_val), average log fold change upon stimulation (avg_logFC), the fraction stimulated (fraction.stim.exp) and unstimulated cells showing expression (fraction.unstim.exp), the Bonferroni-corrected p-value (p_val_adj) and the HGNC name (hgnc_names) are provided. The direction of the effect of the DE genes identified in scRNA-seq data are compared to the direction in bulk RNA-seq data (concordant), unless not significant (n.s.) or not detected (NA). Additionally, the p-value (bulk.p.val) and t-statistic (bulk.p.stat) within the bulk RNA-seq dataset are provided.
(XLSX)

**S2 Table. Pathway enrichment analysis upon *Candida* stimulation in single-cell data.** For each cell type and for all cells together (bulk-like) the differentially expressed (DE) genes were divided into up and down regulated genes. These were all separately used as input for a REAC-TOME pathway analysis. The top10 enriched pathways are shown and p-values were Bonferroni-corrected. Similarly, the 41 DE core genes that were identified in all cell types were used as input for this analysis.
(XLSX)

**S3 Table. Potential cell type-specific receptor-ligand interactions per condition (*Candida* stimulation and RPMI control).** P-values for all tested receptor-ligand interactions for the RPMI control (first tab) and *Candida* stimulated PBMCs (second tab). An explanation of this CellPhoneDB output file can be found at https://www.cellphonedb.org/documentation.
(XLSX)

**S4 Table. Expression quantitative trait loci analysis upon *Candida* stimulation in bulk RNA-seq data.** eQTLs in bulk RNA-seq data in the *Candida*-stimulated (stimulated_eQTLS) or RPMI control (unstimulated_eQTLs) condition. eQTLs for which the effect size before and after *Candida* stimulation changes (response_QTLs_GWAS_annotated). The p-value (PValue), name (SNPName) and chromosome position (SNPChr, SNPChrPos) of the effect SNP, affected gene (ProbeName), alleles to test (SNPType), allele to compare to (AlleleAssessed), Z-score (OverallZScore), gene name (HGNCName), effect size with standard error (Beta.SE), false discovery rate (FDR) and p-value in GWAS on candidemia susceptibility (gwas.pval).
(XLS)

**S5 Table. Underlying numerical data for functional validation experiments.** The underlying numerical data for Figure panels 3E and 3F.
(XLSX)

## Author Contributions

**Conceptualization:** Dylan H. de Vries, Vasiliki Matzaraki, Mihai G. Netea, Lude Franke, Vinod Kumar, Monique G. P. van der Wijst.

**Data curation:** Dylan H. de Vries, Olivier B. Bakker, Harm Brugge, Harm-Jan Westra.

**Formal analysis:** Dylan H. de Vries, Vasiliki Matzaraki, Monique G. P. van der Wijst.

**Funding acquisition:** Lude Franke.

**Investigation:** Dylan H. de Vries.

**Project administration:** Lude Franke, Monique G. P. van der Wijst.

**Software:** Dylan H. de Vries.

**Supervision:** Lude Franke, Vinod Kumar, Monique G. P. van der Wijst.

**Visualization:** Dylan H. de Vries, Harm Brugge, Harm-Jan Westra, Lude Franke.

**Writing – original draft:** Dylan H. de Vries, Vasiliki Matzaraki, Vinod Kumar, Monique G. P. van der Wijst.

**Writing – review & editing:** Olivier B. Bakker, Mihai G. Netea, Lude Franke.

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
