## [Decision Letter · Decision Letter 0]

7 Feb 2020

Dear Dr. van der Wijst,

Thank you very much for resubmitting your manuscript "Integrating GWAS with bulk and single-cell RNA-sequencing reveals a role for LY86 in the anti-Candida host response" for consideration at PLOS Pathogens. As with all papers reviewed by the journal, your manuscript was reviewed by members of the editorial board.  We also managed to secure the same two independent reviewers who commented on the previous submission. As you will read, the reviewers and editors were impressed with the extensive revisions you had made to the manuscript and I am happy to say that, based on these reviews, we are likely to accept this manuscript for publication, providing that you modify the manuscript according to the review recommendations.

One of the reviewers is content with the manuscript as it stands.  The other has raised additional comments, including the suggestion of more experimental work.  It is our view as an editorial board that these additional experiments are not critical for the current manuscript (whilst agreeing with the reviewer that they would undoubtedly add to the story if the data are readily available).  However, we strongly recommend that you include some text modifications to a) integrate the functional assays more with the broader manuscript and b) clarify the interpretation of Figure 3, as suggested. 

We hope that you will not find these revisions too onerous and look forward to receiving your revised manuscript within 30 days. If you anticipate any delay, please let us know the expected resubmission date by replying to this email. 

Sincerely,

Robin Charles May

Associate Editor

PLOS Pathogens

Scott Filler

Section Editor

PLOS Pathogens

Kasturi Haldar

Editor-in-Chief

PLOS Pathogens

orcid.org/0000-0001-5065-158X

Michael Malim

Editor-in-Chief

PLOS Pathogens

orcid.org/0000-0002-7699-2064

Reviewer Comments (if any, and for reference):

Reviewer's Responses to Questions

**Part I - Summary**

Reviewer #1: (No Response)

Reviewer #2: The authors have addressed my comments in full and their study is of general interest to the field and warrants publication in its present form without further revisions.

**Part II – Major Issues: Key Experiments Required for Acceptance**

Reviewer #1: In addressing my previous major concerns, the authors have softened claims about the novelty of the importance of NK cells in candidemia while also providing more evidence for this and have now included promising data validating their claims regarding LY86. In general, the additions and edits to the manuscripts have substantially improved the paper. Most importantly, they show that knockdown of LY86 in THP-1s results in decreased CCR2 expression, causing reduced chemotaxis in response to CCL2 (MCP-1). While these experiments appear carefully done and knockdown and phenotype are robust, they should be connected more to the Candida results in the rest of the paper. Ideally, this would be done by showing migration is decreased in monocytes following stimulation with Candida, and that migration is even lower in AA individuals vs. GG individuals. If such cells are not available for that, I think that at least showing that Candida inhibits migration of THP1s to CCL2 (or citing a reference that does demonstrate that) would be valuable in providing further evidence that your model is correct, possibly revealing a mechanism of Candida immune suppression, and connecting it more to the Candida work in the rest of the paper. Additionally, Tobias Hohl’s lab seems to have provided evidence that your mechanism could be very important in vivo based on Ngo et al 2014, which you cite earlier in your paper, but not for the finding that CCR2-dependent chemotaxis of monocytes is crucial in mouse candidiasis. Again, I think the new RNAi expt is a good addition to the paper. But showing the inhibition of migration by Candida and citing how crucial CCR2+ monocytes are in systemic candidiasis could elevate the importance of your work further.

One other point regarding Figure 3D: The model does not match what is described in the Results. This figure shows a competitive model between LY86 and LY96 for binding of TLR4. It would be helpful to indicate LY86 is also known as MD-1 (and LY96 (MD-2)). More importantly, based on the Results text and a quick perusal of the literature, it would seem that a more likely model involves RP105, which appears to be a divergent TLR with most sequence similarity to TLR4. A model depicting decreased RP105/MD-1 complex signaling being downregulated by Candida, and to a greater extent with the A allele, to decrease CCR2 levels and chemotaxis would seem to be what is supported by their data and the literature.

Reviewer #2: NA

**Part III – Minor Issues: Editorial and Data Presentation Modifications**

Reviewer #1: (No Response)

Reviewer #2: NA

PLOS authors have the option to publish the peer review history of their article (what does this mean?). If published, this will include your full peer review and any attached files.

Reviewer #1: No

Reviewer #2: No
---

## [Editor Report · Decision Letter 1]

19 Feb 2020

Dear Dr. van der Wijst,

We are pleased to inform you that your manuscript 'Integrating GWAS with bulk and single-cell RNA-sequencing reveals a role for LY86 in the anti-Candida host response' has been provisionally accepted for publication in PLOS Pathogens.

Before your manuscript can be formally accepted you will need to complete some formatting changes, which you will receive in a follow up email. A member of our team will be in touch within two working days with a set of requests.

Congratulations and best wishes,

Robin May

Associate Editor

PLOS Pathogens

Scott Filler

Section Editor

PLOS Pathogens

Kasturi Haldar

Editor-in-Chief

PLOS Pathogens

orcid.org/0000-0001-5065-158X

Michael Malim

Editor-in-Chief

PLOS Pathogens

orcid.org/0000-0002-7699-2064

---

## [Editor Report · Acceptance letter]

27 Mar 2020

Dear Dr. van der Wijst,

We are delighted to inform you that your manuscript, "Integrating GWAS with bulk and single-cell RNA-sequencing reveals a role for *LY86* in the anti-*Candida* host response," has been formally accepted for publication in PLOS Pathogens.

Best regards,

Kasturi Haldar

Editor-in-Chief

PLOS Pathogens

orcid.org/0000-0001-5065-158X

Michael Malim

Editor-in-Chief

PLOS Pathogens

orcid.org/0000-0002-7699-2064